# Performance of the IOTA ADNEX Model on Selected Group of Patients with Borderline Ovarian Tumours

**DOI:** 10.3390/medicina56120690

**Published:** 2020-12-11

**Authors:** Adrius Gaurilcikas, Migle Gedgaudaite, Arvydas Cizauskas, Vaida Atstupenaite, Saulius Paskauskas, Dovile Gaurilcikiene, Tomas Birzietis, Daiva Vaitkiene, Ruta Jolanta Nadisauskiene

**Affiliations:** 1Department of Obstetrics and Gynaecology, Medical Academy, Lithuanian University of Health Sciences, LT-44307 Kaunas, Lithuania; migle.gedgaudaite@lsmuni.lt (M.G.); saulius.paskauskas@lsmu.lt (S.P.); divid.d@gmail.com (D.G.); tomas.birzietis@lsmu.lt (T.B.); daiva.vaitkiene@lsmuni.lt (D.V.); ruta.nadisauskiene@lsmuni.lt (R.J.N.); 2Department of Pathological Anatomy, Medical Academy, Lithuanian University of Health Sciences, LT-44307 Kaunas, Lithuania; arvydas.cizauskas@lsmu.lt; 3Department of Radiology, Medical Academy, Lithuanian University of Health Sciences, LT-44307 Kaunas, Lithuania; vaida.atstupenaite@lsmu.lt

**Keywords:** borderline ovarian tumours, ultrasound, transvaginal ultrasound, preoperative assessment, ADNEX

## Abstract

*Background and objectives:* ultrasound is considered to be the primary tool for preoperative assessment of ovarian masses; however, the discrimination of borderline ovarian tumours (BOTs) is challenging, and depends highly on the experience of the sonographer. The Assessment of Different NEoplasias in the adneXa (ADNEX) model is considered to be a valuable diagnostic tool for preoperative assessment of ovarian masses; however, its performance for BOTs has not been widely studied, due to the low prevalence of these tumours. The aim of this study was to evaluate the performance of ADNEX model for preoperative diagnosis of BOTs. *Methods:* retrospective analysis of preoperative ultrasound datasets of patients diagnosed with BOTs on the final histology after performed surgery was done at a tertiary oncogynaecology centre during the period of 2012–2018. *Results:* 85 patients were included in the study. The performance of ADNEX model based on absolute risk (AR) improved with the selection of a more inclusive cut-off value, varying from 47 (60.3%) correctly classified cases of BOTs, with the selected cut-off of 20%, up to 67 (85.9%) correctly classified cases of BOTs with the cut-off value of 3%. When relative risk (RR) was used to classify the tumours, 59 (75.6%) cases were identified correctly. Forty (70.2%) cases of serous and 16 (72.7%) cases of mucinous BOTs were identified when AR with a 10% cut-off value was applied, compared to 44 (77.2%) and 15 (68.2%) cases of serous and mucinous BOTs, correctly classified by RR. The addition of Ca125 improved the performance of ADNEX model for all BOTs in general, and for different subtypes of BOTs. However, the differences were insignificant. *Conclusions:* The International Ovarian Tumour Analysis (IOTA) ADNEX model performs well in discriminating BOTs from other ovarian tumours irrespective of the subtype. The calculation based on RR or AR with the cut-off value of at least 10% should be used when evaluating for BOTs.

## 1. Introduction

Compared to invasive ovarian cancer, borderline ovarian tumours (BOTs) are associated with a significantly better overall survival rate (59.7–99.6%, depending on the stage of the disease) [1,2], but tend to compromise younger patients [1,2,3,4]. The possibility of conservative management and fertility preservation makes preoperative diagnosis of BOTs a very important obstacle [5,6]. Ultrasound is considered to be the primary tool for preoperative assessment of ovarian masses [7,8]; however, the discrimination of BOTs is challenging, and the accuracy of the diagnosis depends highly on the experience of the sonographer [7,9].

In 2014, the International Ovarian Tumour Analysis (IOTA) group delivered the Assessment of Different NEoplasias in the adneXa (ADNEX) model, which is the first risk model to differentiate benign ovarian tumours, BOTs, stage I invasive cancer, stage II–IV invasive ovarian cancer, and secondary metastatic cancer [10].

For the calculation of risk, the ADNEX model uses three clinical variables (patients’ age, Ca125 tumour marker, and whether the ultrasound scan was performed in an oncology center or not) and six ultrasound variables (maximal diameter of the lesion, maximal diameter of the largest solid part, presence of more than 10 locules (Figure 1) in the ovarian lesion, number of papillary projections (Figure 2), presence of acoustic shadows, and ascites) [10].

The ADNEX model is considered a valuable diagnostic tool for preoperative assessment of ovarian masses, especially in the clinical setting with less experienced sonographers. However, its performance for BOTs has still not been widely studied, due to the low prevalence of these tumours.

The aim of this study was to evaluate the performance of ADNEX model for preoperative diagnosis of BOTs.

## 2. Materials and Methods

### 2.1. Ethics

The study was conducted in accordance with the Declaration of Helsinki, and the protocol was approved by the Bioethics Centre of Lithuanian University of Health Sciences (BEC-MF-88).

### 2.2. Design

A retrospective analysis was conducted at Lithuanian University of Health Sciences, Kaunas, Lithuania.

### 2.3. Patients and Ultrasound Collection Parameters

Ultrasound (US) datasets of patients diagnosed with a BOT on the final histology after performed surgery was done, during the period of 2012–2018, were used. Only the patients with primary BOTs were included into the study. The following US parameters were collected: largest diameter of the mass, largest diameter of the solid part of the mass, number of papillary projections, number of locules, presence of ascites, and acoustic shadows.

### 2.4. IOTA ADNEX Model Specifications and Interpretation

The IOTA ADNEX model (web application) was used to calculate the absolute risk (AR), predicting the probability of the mass being a BOT. To evaluate the performance of ADNEX model by AR, different cut-off values (20%, 10%, 5%, and 3%) for the risk of malignancy were tested. The test result was evaluated as “positive” if the risk for BOT was the highest among malignant ovarian tumour groups for each selected cut-off value. The relative risk (RR) was calculated as a proportion of absolute risk and population risk. According to RR, the test was evaluated as “positive” if the risk for a BOT was the highest among benign, invasive, and metastatic ovarian tumour groups. Additionally, the performance of the ADNEX model was evaluated with and without a preoperative level of Ca125 tumour markers (selected cut-off value for malignancy for AR was 10%). Seven cases with missing Ca125 levels were excluded from the calculations.

To evaluate the performance of the ADNEX model in different types of BOTs, the study group was divided into two subgroups, according to the final histology: serous BOTs and mucinous BOTs. Due to the low number, mixed and endometrioid BOTs were excluded from later calculations (*n* = 6). The ARs (10% cut-off value for malignancy has been selected) and RRs, with and without added preoperative levels of Ca125, were calculated for the subgroups of serous and mucinous BOTs, as described previously.

### 2.5. Statistical Analysis

All calculations were made using Statistical Package of Social Sciences, Mac version 26 (SPSS, IBM, Brøndby, Denmark). Continuous variables were described using the median (interval), and categorical variables were reported as frequency and percentage. As data were not distributed normally, a Mann–Whitney U test was used for pairwise comparisons. Pearson’s Chi-square test was used to compare independent categorical variables, while Cochrane’s Q test and McNemar criteria were used to compare the related ones. The statistical significance level was a *p* value less than 0.05.

## 3. Results

### 3.1. Main Characteristics of Study Population

Eighty-five patients were included in the study. The main characteristics of the patients are shown in Table 1.

The main ultrasound features of the BOTs are presented in Table 2 and Table 3.

### 3.2. Performance of ADNEX Model According to Absolute and Relative Risk

The results of ADNEX model-classified cases are presented in Table 4.

The performance of ADNEX model based on absolute risk (AR) depends on the selected cut-off value for the malignancy risk. More encompassing cut-off values allow the model to differentiate BOTs better with an additional number of correctly classified cases, compared to a stricter cut-off value. The results of the ADNEX model performance with the selected different cut-off values are shown in Table 5.

When relative risk (RR) was used to classify the tumours, 59 (75.6%) cases were identified correctly. The comparison of ADNEX performance using RR as a reference and AR with different cut-off values is presented in Table 6.

### 3.3. The Value of Ca125 Marker in ADNEX Model for BOT Classification

According to AR with selected cut-off value of 10%, without Ca125 added, the ADNEX model correctly classified 69.4% of the cases. With Ca125 added to the model, the number of correctly classified cases increased to 71.8%; however, the difference was insignificant (*p* = 0.629).

According to RR, the model without Ca125 classified 72.9% of the BOT cases correctly. An addition of Ca125 increased the performance of the model to 75.6%. However, this change was also insignificant (*p* = 0.453).

### 3.4. The Performance of the ADNEX Model Between Different Histological Subtypes of BOTs

The results of the performance of the ADNEX model for different histological types of BOTs are presented in Table 7.

Ca125 insignificantly increased the number of correctly classified BOT histological subtypes. According to AR, the appliance of Ca125 resulted in additional 7.02% of serous and of 3.1% of mucinous BOTs (*p* = 0.454); according to RR, applying Ca125 resulted in an additional 8.8% of serous and of 9.1% of mucinous BOTs (*p* = 0.453).

## 4. Discussion

The results of our study have demonstrated a good overall performance of the ADNEX model for discriminating BOTs from other ovarian masses. To our knowledge, it is the only study on the performance of the ADNEX model with a representable set of BOT cases carried in an external center.

A more encompassing cut-off for malignancy values, used to differentiate BOTs according to AR, resulted in lower number of BOT cases wrongly classified as benign tumours. However, the difference was only significant between 10% and 20% cut-off values. It was also noted that the number of wrongly classified tumours in malignant categories (stage I and stage II–IV invasive carcinomas) remained the same with all different cut-off values used.

The performance of the ADNEX model based on RR was significantly better compared to AR with a cut-off value of 20%. The better performance of RR was also noted when AR with a 10% cut-off value was used; however, this was not significant. On the other hand, AR with 5% and 3% cut-offs performed better than RR, although these differences were insignificant. It was previously noted that due to the low overall prevalence of some types of ovarian malignancies, such as metastatic lesions, as well as BOTs, RR can be a more informative tool [11]. In our study, we noticed the advantages of RR compared to AR only in the groups with lower cut-off values. Nevertheless, prospective evaluation and bigger sample studies are needed to address this issue.

The value of Ca125 in the diagnosis of BOTs is still controversial. While most of the studies report slightly elevated Ca125 levels, especially in serous BOT groups [2], Eltabbakh et al. and more recent publication by Morroti et al. emphasize that in cases of early invasive cancer and BOTs, the levels of Ca125 marker overlap [3,12]. However, it has been noted that Ca125 could be an independent prognostic factor for peritoneal implants [13,14]. In the ADNEX model, the Ca125 is an optional feature, but the authors state that missing the value of this marker will decrease the accuracy of the model to discriminate between stage II–IV invasive ovarian tumours and other malignancy types [10]. Our results showed the tendency of Ca125 to increase the number of correctly classified BOTs when both AR and RR were employed. It has been also demonstrated with different histological subtypes of BOTs; however, none of these results were statistically significant.

Concerning the ADNEX model’s performance in different subtypes of BOTs, this is the first study addressing the issue. The greyscale ultrasound features of both subtypes in our study agrees with those noted by the other authors: serous BOTs were mainly unilocular, solid tumours with papillary projections [9,15]; while mucinous tumours were described as multilocular cysts, significantly larger than serous tumours [16]. Regardless of the different sonographic features of these subgroups, our results showed a similar performance of the ADNEX model for both histological subtypes, with no significant differences.

Obviously, the retrospective nature of this study is a major limitation; however, we assume that, reviewing the recorded data and protocols of ultrasound examinations adherent to IOTA terminology and measurement technique, we have retrieved credible ultrasound characteristics of adnexal masses.

## 5. Conclusions

The IOTA ADNEX model performs well in discriminating BOTs from other ovarian tumours. The calculation based on RR or AR with the cut-off value of at least 10% should be used when evaluating for BOTs. Ca125 has a tendency to improve the results. The ADNEX model performs equally well in both serous and mucinous BOTs.

## Figures and Tables

**Figure 1 medicina-56-00690-f001:**
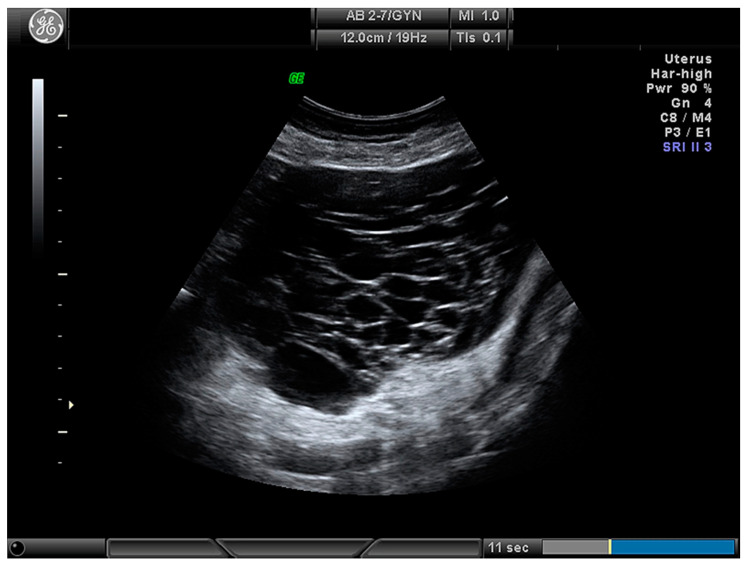
Multilocular ovarian lesion with more than 10 locules.

**Figure 2 medicina-56-00690-f002:**
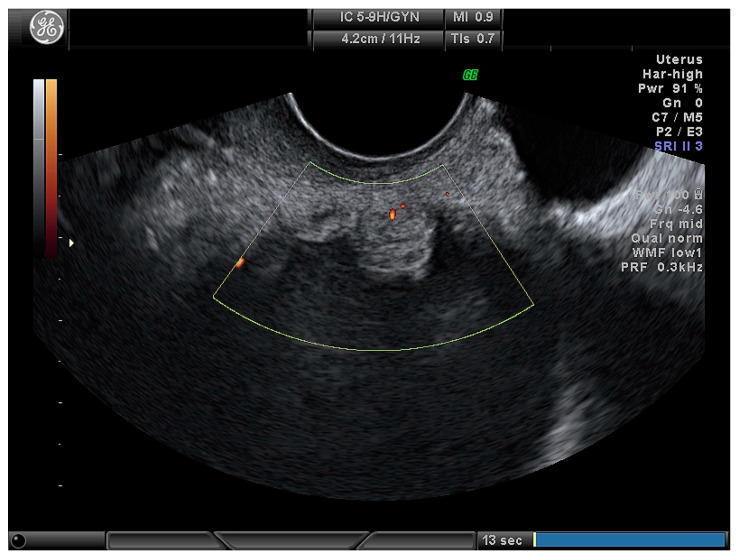
Papillary projection in ovarian lesion.

**Figure 3 medicina-56-00690-f003:**
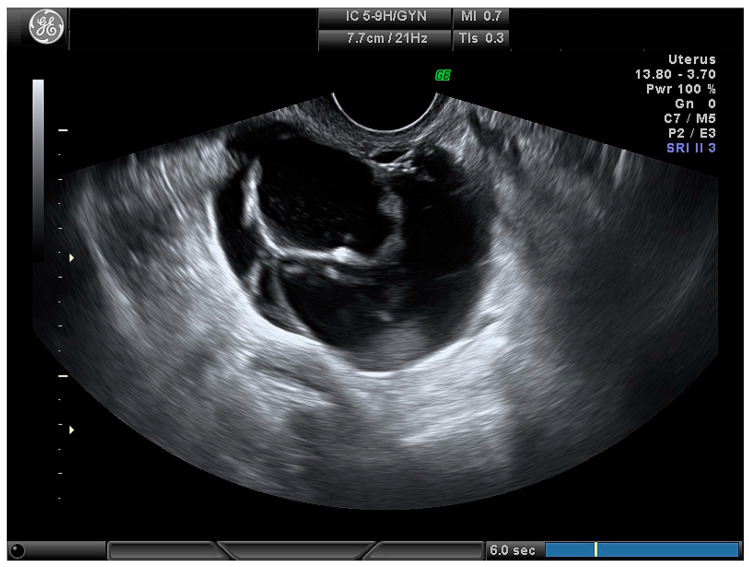
Multilocular cystic ovarian lesion.

**Figure 4 medicina-56-00690-f004:**
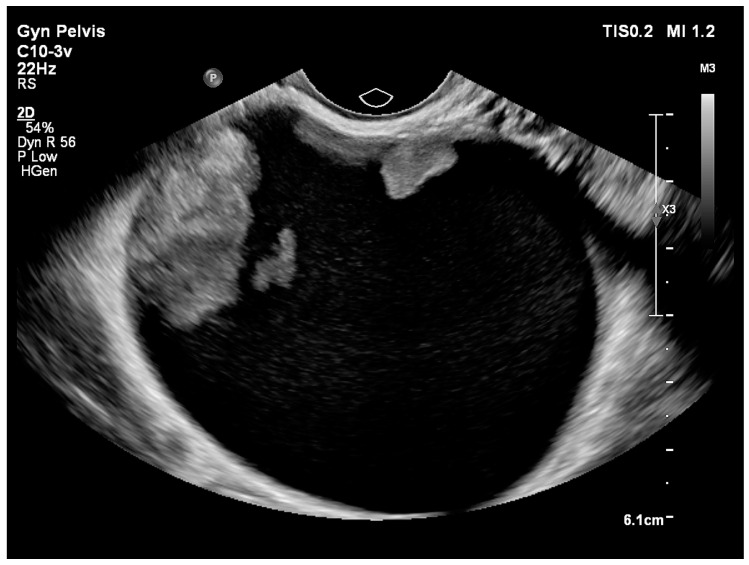
Unilocular cystic–solid ovarian lesion.

**Table 1 medicina-56-00690-t001:** Clinical characteristics of the study population.

Median Age (Range)	46.1 (17–83)
Tumours histology:	
Serous	57 (67.1%)
Mucinous	22 (25.9%)
Other:	
Endometrioid	4 (4.7 %)
Mixed	2 (2.3%)
FIGO Stage:	
I	74 (87.1 %)
II–III	11 (12.9 %)
Median value of Ca125 tumour marker (range), U/mL	72.8 (5.9–918.0) *

* Seven cases were missing a Ca125 value, and were excluded from the analysis when Ca125 was needed.

**Table 2 medicina-56-00690-t002:** Ultrasound features of borderline ovarian tumours (BOTs).

Ultrasound Feature	Median (Interval) or *n* (%)
Maximal diameter of the tumours (mm)	80.0 (20–639)
Maximal diameter of the solid part (mm)	15.0 (3–73)
Type of tumour:	
Cystic (Figure 3)	24 (28.2)
Cystic–solid (Figure 4)	61 (71.8)
Number of locules:	
Unilocular	41 (48.2)
Multiloculcar (total)	44 (51.8)
More than 10 locules	30 (35.3)
Number of papillary projections:	
None	30 (35.3)
1	15 (17.6)
2	5 (5.9)
3	2 (2.4)
More than 3	33 (38.8)
Ascites present	6 (7.1)

None of the cases had acoustic shadows present.

**Table 3 medicina-56-00690-t003:** Ultrasound features of the different BOT histological subtypes.

Ultrasound Feature	Serous BOT	Mucinous BOT	Serous vs. Mucinous
	(*n* = 57)	(*n* = 22)	
	Median (Range) or *n* (%)		
Type of tumour:			*p* < 0.001
Cystic	7 (12.3)	12 (54.5)	
Cystic–solid	50 (87.7)	10 (45.5)	
Number of locules:			*p* = 0.008
Uniloculcar	32 (56.1)	5 (22.7)	
Multilocular	25 (43.9)	17 (77.3)	
More than 10 locules present	14 (24.6)	14 (63.6)	*p* = 0.001
Presence of papillary projections	46 (80.7)	8 (36.4)	*p* < 0.001
Maximal diameter of the tumour (mm)	72.0 (20.5–639.0)	135 (25.7–300)	*p* = 0.003
Maximal diameter of solid part (mm)	15.0 (3.0–73.0)	21.5 (4.0–66.0)	*p* = 0.003

**Table 4 medicina-56-00690-t004:** Performance of the ADNEX (Assessment of Different NEoplasias in the adneXa) model according to relative risk (RR) and absolute risk (AR) with different cut-offs.

ADNEX Result	Number (%) of Cases According to AR with Different Cut-Off Values				Number (%) of Cases According to RR
	3%	5%	10%	20%	
Benign	1 (1.3%)	5 (6.4%)	12 (15.4%)	21 (26.9%)	13 (16.7%)
BOT	67 (85.9%)	63 (80.8%)	56 (71.2%)	47 (60.3%)	59 (75.6%)
Stage I invasive Ca	3 (3.8%)	3 (3.8%)	3 (3.8%)	3 (3.8%)	5 (6.4%)
Stage II–IV invasive Ca	7 (9.0%)	7 (9.0%)	7 (9.0%)	7 (9.0%)	1 (1.3%)

**Table 5 medicina-56-00690-t005:** Changes of ADNEX model performance by AR with different cut-offs.

Cut-Off	Correctly Classified BOT Cases *N* (%)	Absolute Change in Case Number (%)	*p*-Value
20%	47 (60.3%)	−	−
10%	56 (71.2%)	+9 (11.5 %)	0.004
5%	63 (80.8%)	+7 (8.97%)	0.016
3%	67 (85.9%)	+4 (5.1%)	0.375

**Table 6 medicina-56-00690-t006:** Comparison of ADNEX performance between RR as a reference * and AR with different cut-off values.

Cut-Off Value	Difference of the Case Number According to AR	*p*-Value
20%	−12 (15.4%)	<0.001
10%	−3 (3.9%)	0.375
5%	+4 (5.1%)	0.388
3%	+7 (9.0%)	0.118

* reference RR result—59 (75.6%) correctly classified BOT cases.

**Table 7 medicina-56-00690-t007:** ADNEX model performance in serous and mucinous BOT groups.

Without Ca125	Serous BOTs (*n* = 57)	Mucinous BOTs (*n* = 22)	*p*-Value (Serous vs. Mucinous)
AR (cut-off 10%)	40 (70.2%)	16 (72.7%)	0.823
RR	44 (77.2%)	15 (68.2%)	0.409
*p*-value (AR vs. RR)	0.289	1.000	−
**With Ca125**	**Serous BOTs** **(*n* = 53)**	**Mucinous BOTs** **(*n* = 20)**	***p*-Value** **(Serous vs. Mucinous)**
AR (cut-off 10%)	40 (75.5%)	15 (75.0%)	0.967
RR	42 (79.2%)	15 (75.0%)	0.696
*p*-value (AR vs. RR)	0.625	1.000	−

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
