# Peer review of "Performance of the IOTA ADNEX Model on Selected Group of Patients with Borderline Ovarian Tumours"

_medicina, 2020, doi:10.3390/medicina56120690_

Round 1
Reviewer 1 Report
The manuscript was written well and clear.
- For more convenience of the readers, I'd like to suggest all the variables (3 clinical & 6 ultrasonographic) should be described in the introduction section.
- In Table 2 and 3, number of locules were classified unilocular and multilocular. The variable of IOTA-ADNEX is ‘more than 10 locules’, so the number of patients who have more than 10 locules should be informative.
- In Table 6, the number of absolute cases which were corrected classified BOT by RR should be described. It is helpful for the readers..
Author Response
- As a Reviewer suggested, we did some changes and added the description of all the variables in the introduction section. We agree, that this addition makes the text more convenient for the readers.
- The number of lesion with more than 10 locules were added to both tables.
- The absolute number of correctly classified BOTs by RR model is 59. It is stated in the table 6 and in the text above this table. Would it be possible for acknowledged Reviewer to explain this issue in more detailed way?
Reviewer 2 Report
add 3-5 representative images
Author Response
The manuscript was corrected with addition of four pictures that represents the ultrasound features of multilocular, unilocular - solid ovarian lesions. Also a picture with a papillary projection and an ovarian lesion with more than 10 locules.
Reviewer 3 Report
The manuscript by Gaurilcikas et al. presented their work on investigating if the ADNEX model can distinguish different types of malignant ovarian cancer; in particular, they analyzed if this model can correctly identify borderline ovarian tumors (BOTs).
- The introduction is precisely and clearly written, but it lacks a few sentences about the ADNEX model, how it works and how it is used for diagnosis.
- Line 47, it should read adneXa and not adneX, right?
- Lines 115-117, I do not understand where these percentages came from. They are not in any of the tables presented here.
- Line 131, the authors mention that this is one of the few studies, but do not provide references for such studies.
- The authors need to explain why they choose cut off of 10% for evaluation of the different subtypes of BOTs and Ca125. Is 10% the usual cut off?
- Throughout the text, the authors refer to cut off of 3% and 5% as higher and 10% and 20% as lower values. Isn’t it the other way around? I found this very confusing.
Author Response
- The introduction section was corrected and the description of all clinical and ultrasound variables used in the ADNEX model was added. We agree that this correction is helpful and more convenient for the reader.
- Reviewer is right, we corrected the Line 47, it really should read adneXa.
- We would like to ask if it is possible to address this issue more clearly? Maybe the mentioned lines changed during editing process and we cannot clearly understand which percentages are not clear.
- The correction was made. To our knowledge this is the only study that focuses on ADNEX model performance exclusively on borderline ovarian tumours.
- For further evaluation of different subtypes of BOTs and impact of Ca125 we selected the cut-off of 10 % for malignancy risk, because this cut-off value is the proposed one by the developers of the ADNEX model. According to the authors, the cut-off of 10% results in 96.5% sensitivity and 71.3% specificity of this model [5]. According to analysis of our data, the cut-off of 10% results in significantly better results compared to cut-off of 20%. The cut-off values of 3% and 5% increased the number of correctly differentiated cases even further, but the differences were insignificant.
- We strongly agree with the Reviewer on this issue and made corrections throughout the manuscript. We believe that the terms "more encompassing" and "stricter" cut-off values will be more convenient and less confusing for all the readers.